# Antioxidant Capacity of Thistle (*Cirsium japonicum*) in Various Drying Methods and their Protection Effect on Neuronal PC12 cells and *Caenorhabditis elegans*

**DOI:** 10.3390/antiox9030200

**Published:** 2020-02-28

**Authors:** Miran Jang, Kee-Hong Kim, Gun-Hee Kim

**Affiliations:** 1Department of Food Science, Purdue University, West Lafayette, IN 47906, USA; jang126@purdue.edu (M.J.); keehong@purdue.edu (K.-H.K.); 2Plant Resources Research Institute, Duksung Women’s University, Seoul 01370, Korea; 3Department of Food and Nutrition, Duksung Women’s University, Seoul 01370, Korea

**Keywords:** thistle (*Cirsium japonicum*), phenolic compounds, antioxidant, reactive oxygen species (ROS), PC 12 cells, *Caenorhabditis elegans*

## Abstract

The aim of this study was, firstly, to evaluate the phenol profile of thistle (*Cirsium japonicum*, CJ) by High performance liquid chromatography-electrospray ionization–mass spectrometry (HPLC–ESI–MS), dried by different methods (90 °C hot-air, 70 °C hot-air, shade-, and freeze-drying). Secondly, we aimed to evaluate the relationship between phenolic compounds content and antioxidant properties. CJ contained chlorogenic acid, linarin, and pectolinarin. Total phenolic contents of CJ significantly decreased under hot-air-drying condition, especially chlorogenic acid contents in CJ have been reduced by 85% and 60% for 90 °C and 70 °C hot-air-drying, respectively. We evaluated the protective effect on adrenal pheochromocytoma (PC12) cells and *Caenorhabditis elegans* using shade-dried CJ, which has the largest phenolic contents and the strongest antioxidant property. CJ-treated PC 12 cells dose-dependently exhibited the protective effects against reactive oxygen species (ROS), while cell viability increases, lactate dehydrogenase release decreases, and ROS formation decreases. Furthermore, CJ has also shown protection against ROS in *C. elegans.* Consequently, CJ contributed to lifespan extension under ROS stress without influencing the physiological growth.

## 1. Introduction

People in North America have used the roots of thistles to treat wounds, boils, piles, and stomachache [1]. In Asia, traditionally, the aerial part of thistles has been used as food and medicine after boiling for purpose of cooling blood and maintaining hemostasis [2]. Thistle is characterized by leaves that contain prickles around the edges. These plants include genera Cardueae, Carduus, Onopordum, and Cirsium. Among thistles, *Cirsium japonicum* (CJ) belongs to the Composiae family and grows most commonly in Asia.

The major compounds contained in CJ include chlorogenic acid, linarin, pectolinarin, apigenin, luteolin, acacetin, hispidulin, silymarin, cirsmarin, and cirsmaritin [3,4,5,6]. CJ extract has been previously reported for its antioxidant and anti-inflammatory effect; however, these reports are mainly on hepatocellular and/or liver target efficacies [5,7]. Recently, research regarding CJ is underway on the preventive efficacies on age-related brain and neuronal degeneration such as anti-menopausal [8,9], gamma-aminobutyric acid (GABA) level modulate [10,11,12], and neuronal protective effects [4,13].

Age-related neural diseases and degeneration are known to be mainly caused by the excessive presence of reactive oxygen species (ROS) [14]. Several factors may contribute to oxidative stress in aged brains. First, mitochondrial dysfunction found in aged brains may exaggerate the generation of ROS [15]. Second, amyloid-beta peptides have been reported to generate ROS in the presence of metal ions such as Fe^2+^ and Cu^2+^ in the brains of Alzheimer’s disease patients [14,15]. Protein and DNA damage and lipid peroxidation caused by ROS in the brain, which is particularly high in lipids, may be manifested in the form of physical, chemical, or functional changes that may ultimately cause neurodegeneration [16]. Oxidative stress has been considered as a therapeutic target for the treatment of neuronal degeneration [17]. Based on the theory that oxidative stress causes neurodegenerative disorders, various physiological studies highlighted the therapeutic potential of phenolic compounds as antioxidants: radical scavengers and ion chelators [14,15,18,19].

To determine in vivo functional effect on bio-active components, researchers have newly introduced and employed *Caenorhabditis elegans*. In studies on aging, in particular, *C. elegans* is useful for investigation screening due to its short lifespan of less than a month and its large brood scale. Additionally, at least 83% of protein sequences of *C. elegans* are homologous to human genes and have been matched with human gene transcripts [20,21].

On the other hand, we adopted drying operations to prevent the deterioration of the quality caused by moisture so that it could be maintained the functional ingredients such as polyphenol and flavonoid in the plants during storage. Since drying of plants inhibits microbial growth and prevents biochemical changes [22], the drying process plays an important role to reduce storage volume and extend their shelf life [23,24]. In dried plants, enzymes such as polyphenol oxidases were inactivated due to decreased water activity, which thus enables retaining internal quality such as phenolic contents and antioxidant properties of the extract [22]. However, during drying processes, sensitive compounds are broken down if the compounds are exposed to specific stress conditions such as heat, air, and light.

We conducted high-performance liquid chromatography-electrospray ionization tandem mass spectrometry (HPLC-ESI-MS) analysis to establish marker phenolic compounds and evaluated the validity of the compounds for the purpose of preventing the loss of active compounds and negative changes according to drying conditions of CJ. Therefore, in this study, we identified phenolic components and analyzed the change in phenolic contents based on differing drying methods of CJ. Then, we evaluated whether the change of phenolic contents influences antioxidant properties. To investigate the beneficial effect of CJ on aging, we treated CJ extract into oxidative stress-induced PC12 cells with CJ extracts. We confirmed the role of these extracts in protection against ROS using the in vivo model, *C. elegans*, which also positively influences the lifespan of *C. elegans.*

## 2. Materials and Methods

### 2.1. Thistle Preparation and Drying

Thistle (*Cirsium japonicum*, CJ) was collected at Pocheon (37°53’41.7” N 127°12’01.3” E), South Korea. The collected plants were visually authenticated by the Korean National Arboretum. The collected CJ was transported to the laboratory within 24 h. We removed flowers and seeds from the aerial parts of CJ and washed them in cold tap water twice. CJ was drained and cut to 10–15 cm before drying. To value the relative effect of the drying methods, we measured the weight of fresh CJ and dried CJ using a moisture balance (MB45 moisture analyzer, Ohaus, Parsippany, NJ, USA). We assumed that once the ratio of the initial-final CJ weight after each drying process reached the measured value in the water analyzer, the CJ was completely dried. (a) 90 °C hot-air-drying was conducted for 30–36 h and (b) 70 °C hot-air-drying was conducted for 48–54 h in an hot-air circulation oven (VB-200DL, Viosionbiotech, Gyeonggi, Korea), (c) Shade-drying was conducted at ambient temperature (25.0 ± 1.0 °C) in the dark and airy room for 2 weeks, and (d) Freeze-drying was conducted in a freeze-dryer (Ilshinbiobase, Gyeonggi, Korea) at 0.5 mbar and −80 °C in for 48 h. All prepared CJ was extracted by reflux extraction at 60 °C for 6 h using 70% (*v/v*) alcohol. The extracts were combined and evaporated using an evaporator (EYELA, Tokyo Rikakikai Co., Tokyo, Japan) under 50 °C. The CJ extract was dissolved in dimethyl sulfoxide (DMSO) at 10 mg/mL as a stock solution and stored at -80°C until analysis.

### 2.2. HPLC and LC-ESI-MS Analysis

The phenolic compounds from CJ were analyzed using HPLC–ESI–MS. The analytical condition is adopted from Jang et al. [4] and indicated in Table 1. Quantification was performed by external calibration using standards. Calibration curves for the standard phenolic compounds were in the range 5–100 μg/mL in which the linearity of the response was given and linear a correlation coefficient ≥0.98. For identification through the molecular weight, separated individual peaks were characterized using MS analysis. As the maker components of CJ, chlorogenic acid and phosphoric acid was purchased from Sigma-Aldrich (St. Louis, MO, USA) and linarin and pectolinarin were isolated with column chromatography and provided by the Cheil Jedang Functional Food Research & Development Center (Seoul, Korea). Acetonitrile (ACN) of HPLC grade were purchased from Millipore (Bedford, MA, USA).

### 2.3. Antioxidant Capacities

#### 2.3.1. DPPH Assay

The 2,2-diphenyl-1-picrylhydrazyl (DPPH) assay modified and used the method by Jang et al. [25]. Briefly, same volume of sample and DPPH solution (0.2 mM) were mixed and placed for 30 min at room temperature. The reaction mixtures were recorded at 517 nm using the microplate reader (SpectraMax M2, Molecular Devices, CA, USA). DPPH radical scavenging activity was calculated using the following formula:

DPPH radical scavenging activity (%) = [1 − (sample O.D./blank O.D.)] × 100

The values of antioxidant capacity were calculated using ascorbic acid (AA) calibration curve and expressed as micrograms of AA per gram of sample dry weight (DW).

#### 2.3.2. ABTS Assay

To measure the scavenging capacity against 2,2’-azino-bis (3-ethylbenzothiazoline-6-sulphonic acid) (ABTS), we modified and used the method by Gullon et al. [26]. To produce ABTS cation, the mixture containing 7.0 mM ABTS (in 20 mM sodium acetate buffer, pH 4.5) and 2.45 mM potassium persulfate kept in dark for overnight. ABTS solution was diluted to an absorbance of 0.7 ± 0.01 at 734 nm. 50 μL of samples react with 950 μL of ABTS solution and then, was placed for 10 min at room temperature and after was recorded an absorbance at 734 nm. ABTS radical scavenging activity was calculated using the following formula:

ABTS radical scavenging activity (%) = [1 − (sample O.D./blank O.D.)] × 100

The values of antioxidant capacity were calculated using ascorbic acid (AA) calibration curve and expressed as micrograms of AA per gram of sample dry weight (DW).

#### 2.3.3. ORAC Assay

The oxygen radical absorbance capacity (ORAC) assay was conducted with the method by Jang et al. [25] Dilutions of all reagent were prepared in 75 mM phosphate buffer (pH 7.4) at 37 °C. A 100 μL sample and 50 μL of 8.16 × 10^−5^ mM fluorescein were dispensed into 96-well plates. Then, 50 μL of 153 mM AAPH, which was preincubated at 37 °C for 15 min, were subsequently dispensed into each well, and the plate was immediately measured. The reader was programmed to record the fluorescence every 3 min for 60 min at emission and excitation wavelengths of 538 nm and 485 nm, respectively. ORAC values were calculated using the following formula:

AUC (area under the curve) = 1 + (f1/f0) + (f2/f0) + (f3/f0) + ·········· + (f19/f0) + (f20/f0)

Relative ORAC value = [(AUC sample − AUC blank)/(AUC ascorbic acid − AUC blank)]

The values of antioxidant capacity were calculated using ascorbic acid (AA) calibration curve and expressed as micrograms of AA per gram of sample dry weight (DW).

#### 2.3.4. FRAP Assay

Ferric reducing antioxidant power (FRAP) assay was followed according to the method by Gullon et al. [26]. FRAP reagent was prepared with a 10:1:1 (v:v:v) mixture of three solutions; 300 mM acetate buffer (pH 3.6), 20 mM ferric chloride, and 10 mM 4,6-tripryridyls-triazine (TPTZ) made up in 40 mM HCl. The FRAP solution was placed at 37 °C for 15 min. 50 μL of sample and the prepared FRAP solution were mixed then placed in the dark for 30 min at room temperature. The reaction mixtures were recorded at 593 nm.

The values of antioxidant capacity were calculated using ascorbic acid (AA) calibration curve and expressed as micrograms of AA per gram of sample dry weight (DW).

#### 2.3.5. FICA Assay

To determine the ferrous ion chelating ability (FICA) [26], 30 μL of samples were mixed with 250 μL of 100 mM sodium acetate buffer (pH 4.9) and 30 μL of 2 mM aqueous FeCl_2_ solution. After 10 min reaction, 12.5 μL of an aqueous 10 mM ferrozine [3-(2-pyridyl)-5,6-diphenyl-1,2,4-triazine-4,4”-disulfonic acid] solution was added. The reaction mixtures were recorded at 562 nm.

The FICA values were expressed as micrograms of Ethylenediaminetetraacetic acid (EDTA) per gram of sample dry weight. The calibration curve of EDTA showed with a correlation coefficient of ≥0.98.

### 2.4. Neuronal Cell Protective Effects on Oxidative Stress

#### 2.4.1. Cell Culture

PC12 cell line (KCLB 21721), derived from rat pheochromocytoma, was obtained from the Korea Cell Line Bank (Seoul, Korea). All experimental methods on PC12 cells were adapted from Jang et al. [4] and modified. We treated nerve growth factor (NGF) into PC12 cells for the differentiation into neuronal phenotype. PC12 cells were grown in roswell park memorial institute (RPMI) 1640 supplemented with 5% (*v/v*) fetal bovine serum (FBS), 100 units/mL of penicillin, and 100 μg/mL of streptomycin at 37 °C in a humidified atmosphere of 10% CO_2_/90% air. The medium was changed every other day and the cells were incubated for 7 days, a period that could spread out to an appropriate density (approximately 75%–80%). For the experiments using PC12 cells, RPMI 1640 medium and FBS were obtained from Welgene (Daegu, Korea), hydrogen peroxide (H_2_O_2_, 30% (w/w) in H_2_O, contains stabilizer) was obtained from Sigma-Aldrich (St. Louis, MO, USA) and used within six months of the purchase, and other unspecified reagents and chemicals were from Sigma-Aldrich (St. Louis, MO, USA). We used the final concentration of CJ in a range of 0–50 μg extract per mL appropriately diluted for each experiment.

#### 2.4.2. Cell Viability

PC12 cells were plated at a density of 2 × 10^5^ cells/well in 96-well plates and the cell viability was measured using 3-(4,5-dimethylthiazol-2-yl)-2,5-diphenyl tetrazolium bromide (MTT) assay. 200 μL of medium containing CJ extracts were added to the wells and incubated for 24 h before treatment H_2_O_2_. The cells were induced oxidative toxicity with H_2_O_2_ (final concentration 250 μM) for 3 h, and were then incubated for 2 h with MTT (30 μL of a 5 mg/mL stock solution in phosphate-buffered saline, PBS). DMSO (≤0.2% (*v/v*)), which is used when treating the CJ extract, was used for the control group.

MTT, a water-soluble yellow tetrazole, was reduced to purple formazan in alive PC12 cells. We removed an upper layer solution to measure the amount of purple-colored formazan after the MTT reaction and then dissolved the crystals in DMSO. The absorbance of purple formazan dissolving solution can be quantified by measuring at 570 nm. The viability of PC12 cells in each treatment was presented as a relative percentage of the control cells with only DMSO (≤0.2% (*v/v*)) without CJ treatment.

#### 2.4.3. Cell Membrane Damage

The plasma membrane damage of PC12 cells was determined by the levels of Lactate dehydrogenase (LDH) released into the medium. Similar to the MTT assay, CJ extract was added into the cells and incubated for 24 h. After pre-incubation, H_2_O_2_ (250 μM) was treated and the cells were incubated for 3 h. To spectrophotometrically determine, the amount of released LDH was measured using the LDH kit (Sigma-Aldrich Chemical Co., St. Louis, MO, USA). The release of LDH of PC12 cells in each treatment was presented as a relative percentage of the control cells with only DMSO (≤0.2% (*v/v*)) without CJ treatment.

#### 2.4.4. Determination of ROS Level

The levels of cellular oxidative stress were determined using the fluorescent probe 2’,7’-Dichlorodihydrofluorescein diacetate (DCF-DA). The PC12 cells were pretreated with CJ extract for 24 h, and after, the cells were treated with or without H_2_O_2_ (250 μM) for 3 h. Then, the cells were incubated in the presence of 20 μM DCF-DA (dissolved in PBS) for 30 min. Fluorescence was finally quantified equipped with 485 nm excitation and 535 nm emission filters. The oxidative stress levels were expressed as a relative percentage of the control cells with only DMSO (≤0.2% (*v/v*)) without CJ treatment.

### 2.5. The Protective Effects on Oxidative Stress in vivo C. elegans

#### 2.5.1. Worm Culture

The strain of *C. elegans* (wild-type N2) and their food, OP50 (*Escherichia coli*) in the study were provided by the Caenorhabditis Genetics Center (CGC, University of Minnesota, Minneapolis, MN, USA). Worms were grown on nematode growth media (NGM) plates with OP50 at 20 °C. All solutions that are required to grow *C. elegans* such as NGM, M9 buffer, and S-complete were prepared according to the worm book [27]. Bleach (6% hypochlorite, Clorox Company, Oakland, CA, USA) was used to corrode the torso of the adult nematodes torso for obtaining the embryos during synchronizing. The chemicals used for the *C. elegans* experiments were from Thermo Fisher Scientific, Inc., (Pittsburgh, PA, USA), unless indicated otherwise. All experimental methods for *C. elegans* were adapted from Shen et al. [28] and modified.

#### 2.5.2. Population Rate, Reproduction Rate, and Pumping Rate Assay

CJ was added to the synchronized L1 worms in 12-well plates. The number of worms at each developmental stage after 50 h was counted to calculate the growth rate. Worms were transferred to fresh NGM plates every day during the reproduction periods, and the eggs left were allowed to hatch and grow to the L4 stage before counting the number of progenies of each worm.

For the pumping rate, the number of pharyngeal contractions of 15 randomly selected nematodes was counted under the microscope for 30 s.

#### 2.5.3. Determination of ROS Level in *C. elegans*

The levels of cellular oxidative stress were determined using the fluorescent probe DCF-DA and modified for *C. elegans*. After 96 h treatment of CJ extracts, worms were washed three times with M9 buffer. Then, 50 worms were transferred to 50 μL of M9 buffer in a 96-well plate, and then mixed with 50 μL of fresh 100 mM DCF-DA solution in M9 buffer. A well containing 50 mM DCF-DA solution in M9 buffer with/without 3 mM paraquat (paraQ) was regarded as background. Fluorescence was finally quantified equipped with 485 nm excitation and 535 nm emission filters. The oxidative stress levels were expressed as a relative percentage of the control cells without any treatment.

#### 2.5.4. Lifespan Assay

For the lifespan assay, 40 h aged nematodes were transferred in transwell-24 well plate and treated with CJ during the rest of the lifespan study. FUdR (120 mM) was also added to stop producing progeny and freeze original population. All lifespan assays were performed at 20 °C and the medium was changed every two days. Worm survive number was recorded every other day until all worms died. The day that the treatment began is considered as day 0. Online application of survival analysis (OASIS; sbi.postech.ac.kr/oasis) was used for statistical analysis.

### 2.6. Statistical Analysis

The results are expressed as the mean ± standard deviation. Statistical analyses were conducted using SPSS ver. 19.0 statistical analysis software (Chicago, IL, USA). Comparison between groups were assessed by ANOVA (Duncan’s multiple range tests) based on 95% (*p* < 0.05). Student’s t-test was used to assess differences between groups. Statistical significance of t-test has been set at *p* < 0.05 and *p* < 0.01.

## 3. Results and Discussion

### 3.1. The Effects of Different Drying Methods on the Change Of Phenolic Contents in CJ Extract

The profiling illustrations of phenolic compounds of CJ by four different drying conditions (90 °C hot-air-, 70 °C hot-air-, shade-, and freeze-drying) are shown in Figure 1. CJ has shown three major phenolic compounds; chlorogenic acid, linarin, and pectolinarin (Figure 1). Chlorogenic acid (3.51–22.14 mg/g DW) was a predominant phenolic compound in CJ extract. The contents of linarin and pectolinarin in CJ have shown 3.02–6.96 mg/g DW and 0.68–0.96 mg/g DW, respectively (Table 2). The sum of the contents of the three phenolic compounds was indicated in a decreasing order: SCJ > FCJ > 70HCJ > 90HCJ. In particular, chlorogenic acid is sensitive to heat drying compared to linarin and pectolinarin as shown in the result of 84% reduction after hot-air-drying at 90 °C treatment. Chlorogenic acid content declining drastically in CJ affected the total amount of phenolic compounds. Like our result, another example of reduction of chlorogenic acid due to heat could be an outbreak in coffee roasting, especially under higher temperature roasting or longer time roasting [29,30]. In contrast with our results, however, thermal cooking such as grilling, baking, microwaving, and roasting induced an increase of chlorogenic acid in plants [31,32]. In another example, pre-thermal treatment for a short time even at high-temperature before full drying and extraction has shown a positive effect on the increase in phenolic contents [29,30,31,32,33]. Normally, when heat is applied to plants, it ruptures the tissue, leading to an increase in the bioactive ingredient elution. However, the constant heating after the rupture of the tissue induces the degradation of active ingredients.

### 3.2. The Effects Of Different Drying Methods on the Antioxidant Capacities of CJ Extract

To evaluate the antioxidant potencies of CJ extracts, DPPH, ABTS, ORAC, FRAP, and FICA assay were employed (Figure 2). In the radical scavenging test, DPPH, ABTS, and ORAC values of CJ were got lowered in a decreasing order: SCJ > FCJ > 70HCJ > 90HCJ. As the results obtained from the radical scavenging assay indicated, CJ showed similar results on ferric ion reduction and ferrous ion chelating. The antioxidant activities showed the same pattern of the results on their phenolic contents. Antioxidant activity has a positive correlation with phenolic contents [4,25]. Based on these results, the temperatures above 70 °C applied until a complete dry state could reduce the phenolic content consequently reducing antioxidant properties. In the previous research, DPPH radical scavenging activity and levels of polyphenolic compounds in the air-dried mulberry leaves at ≤60 °C did not significantly differ with the freeze-dried mulberry leaves, whereas both values of the air-dried mulberry leaves at ≥70 °C decreased significantly [34], which corresponds with our result. In addition, chlorogenic acid was shown to be strongly influenced by heat treatment and to have the largest contribution to radical scavenging activity [34]. Our results also confirmed that chlorogenic acid, the most potent antioxidant, degrades at temperatures above 70 °C. Meanwhile, decline in total phenolic content and antioxidant activities are often accompanied by loss of other bioactive properties. Since the collapse of the antioxidative defense system is considered to affect aging, a decrease in antioxidant activity could negatively affect age-related damage [35].

We anticipated that phenolic substances, an antioxidant in CJ, would have a positive effect on the physiological activities related to aging. Thus, we induced oxidative damage on neuronal phenotype PC 12 cells (in vitro cellular) and *C. elegans* (in vivo) to confirm the effect of CJ extract on regression caused by aging.

### 3.3. PC 12 Cell Protective Effects of CJ Extract Against ROS

Mitochondria may be the most sensitive primary targets of oxidative damage in neuronal cells [36]. The effect of CJ on the cell viability based on mitochondrial respiration of the living cells was determined by an MTT assay. The cytotoxicity by CJ extracts was not observed in the concentration range of 10–100 μg/mL. The cell viability showed a general pattern of steadily increasing until 50 μg/mL then decreasing from 50 to 100 μg/mL (Figure 3A). From the result, 25 and 50 μg/mL of CJ was selected as the optimal concentration for subsequent experiments for PC12 cell protection. To determine whether there is a significant difference in the effects of various CJ extracts on cell viability, four different CJ extracts were tested.

After incubation with H_2_O_2_ (250 μM) for 3 h, the cell viability was decreased by 44%. However, oxidative stress-induced cytotoxicity was dose-dependently inhibited in CJ-pretreated cells (Figure 3B). The protective effects of CJ on ROS showed in a decreasing order: SCJ > FCJ > 70HCJ > 90HCJ. Such pattern supported the results of phenolic contents and antioxidant capacities (Table 2 and Figure 2). In order to confirm the main contributors of the effect, we prepared a mixture of the following three phenolic compounds: chlorogenic acid, linarin, and pectolinarin. Each mixture was prepared as the concentration (shown in Table 2) of three compounds present in original CJ extract derived from different drying processes. Each mixture had similar effects as its corresponding CJ extract. The results suggest that PC12 cell death by oxidative stress was suppressed by pretreatment with phenolic compounds from CJ extract. SCJ was considered to be most optimal and was used for subsequent experiments based on the results of phenolic contents, antioxidant capacities, and cell viability.

Since LDH is released from apoptotic cells, the LDH release levels can determine what percentage of cells is dead [4]. The H_2_O_2_-induced cytotoxicity was assessed using the LDH assay to examine the protective effect of CJ, which could decrease cell membrane damage. Inhibition rates of CJ against H_2_O_2_-induced membrane damage were shown in Figure 3C. 250 μM H_2_O_2_ caused an increase in LDH release into the media by 61% compared to the control. However, pretreatment with 50 and 25 μg/mL CJ extracts inhibited LDH release by 21% and 29%, respectively. Consistent with the cell viability result, the mixture of three phenolic compounds from 50 μg/mL CJ extract has shown a similar effect to the original CJ extract.

The antioxidant phenolics of CJ extract has been proven to have a cellular protective effect against oxidative damage. Thus, further investigation was needed on whether the CJ extract also has a positive effect on the inhibition of ROS generation. The DCFH-DA assay was used to examine the intracellular formation of ROS in PC12 cells. The DCFH-DA probe can be permeated freely through the cell membrane and is oxidized to a highly fluorescent substance (DCF) when it reacts with ROS [37].

When PC12 cells were maintained under oxidative stress with H_2_O_2_ for 3 h, it resulted in a 133% increase in the ROS levels compared to control (Figure 3D). Pretreatment of CJ extracts in PC12 cells significantly prevented intracellular ROS accumulation in comparison to the H_2_O_2_ treated control (*p* < 0.05). The inhibition of ROS formation showed that the pattern of neuronal cell protection was similar to the LDH assay. These findings suggested that the antioxidant properties of CJ decrease the H_2_O_2_-induced oxidative stress in PC 12 cells and protect the oxidative stress-induced neuronal damage.

### 3.4. Effect of CJ Extract on Growth and Protective Effect Against Oxidative Stress of C. elegans

The growth rate and progeny production on treatments were evaluated to investigate whether CJ extract has any influences on the growth functions of *C. elegans*. As shown in Figure 4A, 18–22% of the worms were at the late L4 stage after 50 h of incubation, and 60–65% reached the young adult stage for all the three groups. This suggests that there was no difference in the growth rate of *C. elegans* between the control and two experimental doses of CJ extracts, which means CJ extract has no influence on normal growth. Additionally, the progeny production result showed that exposure to both 25 and 50 μg/mL of CJ extracts did not significantly alter the population size of *C. elegans* (Figure 4B). To total the results, our data demonstrated that 25 and 50 μg/mL of CJ extracts did not contribute to the normal physiological growth of *C. elegans*.

We next analyzed the pumping rate as an aging-related parameter to evaluate the effect of CJ extract during aging. The pumping speed decreases steadily as the nematodes get older [21,26]. As seen in the control group, the pumping rate on day 14 was reduced by 81% compared to day 1 (Figure 4C). In CJ extract-treated *C. elegans* at both 50 and 100 μg/mL significantly delayed the decline of the pumping rate compared to control (Figure 4C).

As mentioned above, ROS is considerably contributing to aging, and treating ROS scavenger could delay aging. We employed *C. elegans* to confirm whether the CJ effect is shown in vivo. We treated CJ extract, a potential ROS scavenger, then measured ROS accumulation in *C. elegans* under normal growth and oxidative stress-induced condition.

Under normal growth condition, treatment with 50 and 100 μg/mL CJ extract for 96 h greatly reduced intracellular ROS levels by 37% and 39%, respectively, compared to the control (Figure 4D). On the other side, treatment with 50 and 100 μg/mL CJ extract for 96 h greatly reduced intracellular ROS levels by 12% and 13%, respectively, compared to the control under oxidative stress condition induced by ParaQ (3 mM) (Figure 4D).

### 3.5. Positive Effect of CJ Extract on the Lifespan of C. elegans

Based on the outcomes which decrease age-related changes after CJ treatment, we expected CJ extract to be able to extend the lifespan of nematodes. Thus, we determined the effect of CJ extract on the lifespan under normal growth condition and ROS-induced stress condition. The final lifespan of CJ treated nematodes increased by two days compared to the untreated group under normal conditions (no statistical difference) (Figure 5). Meanwhile, treating CJ significantly increased the lifespan of nematodes under the oxidative stress condition induced by 3 mM ParaQ (Figure 5).

As we predicted, the antioxidant phenolic compounds in CJ showed protective effects in oxidative stressed PC12 cells and *C. elegans.* The current results showed that CJ extract diminished the ROS level, which is consistent with the previous study showing the fact that CJ reduced ROS level [4,7,38].

To conclude, we found that the phenolic compounds in CJ were degraded by heat drying over 70 °C; especially chlorogenic acid decreased drastically. The antioxidant phenolic contents were proportional to the antioxidant capacity and showed protection against oxidative stress in PC12 cells and *C. elegans*. We proved that their antioxidant properties, ultimately, showed improvements in aging-related changes and life extension effects in nematodes.

The role of individual phenolic compounds of CJ in a mechanism through the activation of antioxidant signaling pathways related to aging is needed to be further investigated. In particular, since the effect of CJ related to antioxidants is considered coming from chlorogenic acid, the deeper study on the effect of chlorogenic acid is required. Phenolic compounds, and mainly chlorogenic acid, contained in CJ are expected to be used as potential antioxidant nutraceuticals for delaying aging.

## Figures and Tables

**Figure 1 antioxidants-09-00200-f001:**
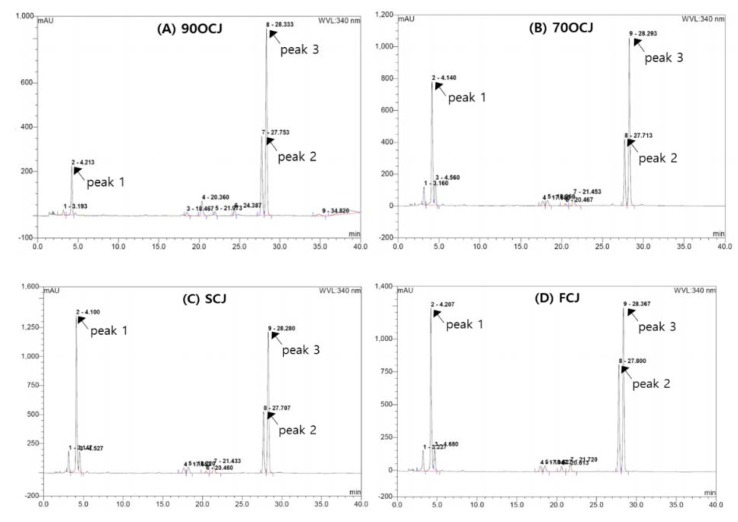
High performance chromatography profiles on phenolic compounds of CJ extracts that are prepared different drying methods (90HCJ: hot-air-dried *C. japonicum* at 90 °C, 70HCJ: hot-air-dried *C. japonicum* at 70 °C, SCJ: shade-dried *C. japonicum* and FCJ: freeze-dried *C. japonicum*. peak 1: chlorogenic acid, peak 2: linarin and peak 3: pectolinarin.).

**Figure 2 antioxidants-09-00200-f002:**
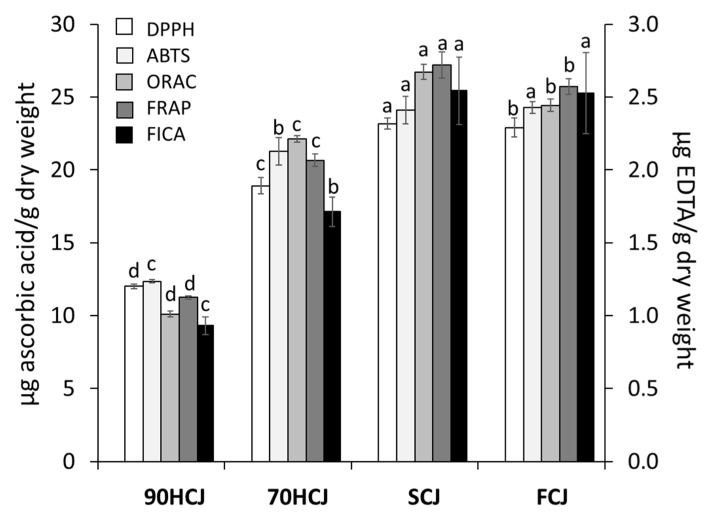
Antioxidant properties of CJ according to four different drying methods drying methods: 2,2-diphenyl-1-picrylhydrazyl (DPPH), 2,2’-azino-bis (3-ethylbenzothiazoline-6-sulphonic acid) (ABTS), oxygen radical absorbance capacity (ORAC), and ferric reducing antioxidant power (FRAP). The values are mean ± SD of three individual experimental results (n = 3). The letters presented above the same color bars represent statistically significant differences (*p* < 0.05). The values of antioxidant capacity were calculated using ascorbic acid (AA) calibration curve and expressed as micrograms of AA per gram of sample dry weight (DW) in DPPH, ABTS, ORAC, and FRAP assay. The ferrous ion chelating (FIC) values were expressed as micrograms of Ethylenediaminetetraacetic acid (EDTA) per gram of sample dry weight. 90HCJ: hot-air-dried *C. japonicum* at 90 °C, 70HCJ: hot-air-dried *C. japonicum* at 70 °C, SCJ: shade-dried *C. japonicum* and FCJ: freeze-dried *C. japonicum*.

**Figure 3 antioxidants-09-00200-f003:**
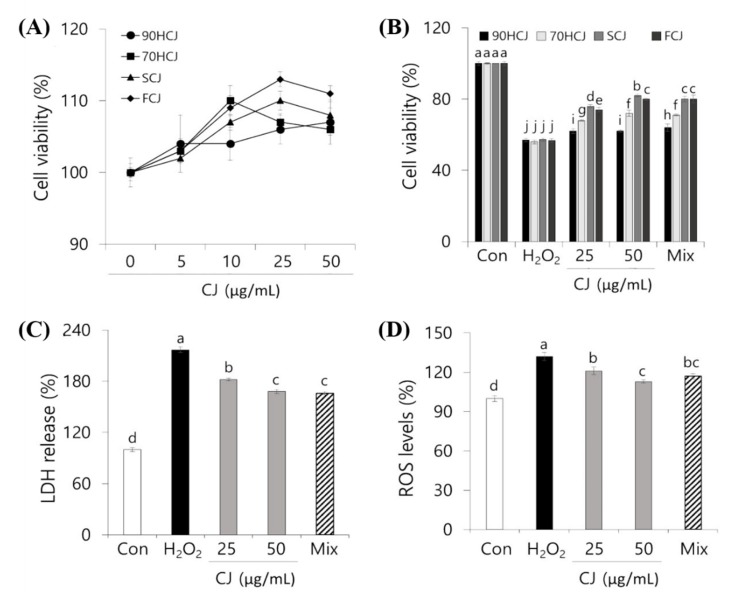
Protective effect of CJ on PC12 cell against oxidative stress. (**A**) Effect of CJ extract on viability of PC12 cells by MTT assay and (**B**) Effect of CJ extract on the viability of PC12 cells that were induced oxidizing damage with H_2_O_2_ (**C**) Effect of CJ extract on H_2_O_2_-induced membrane damage in PC12 cells. (**D**) Effect of CJ extract on ROS generation in PC 12 cells damaged by H_2_O_2_. The values are mean ± SD of three individual experimental results (n = 3). (B) The letters presented above the same color bars represent statistically significant differences (*p* < 0.05). (**C**) and (**D**) The letters presented above the bars represent statistically significant differences (*p* < 0.05). 90HCJ: hot-air-dried *C. japonicum* at 90 °C, 70HCJ: hot-air-dried *C. japonicum* at 70 °C, SCJ: shade-dried *C. japonicum* and FCJ: freeze-dried *C. japonicum.*

**Figure 4 antioxidants-09-00200-f004:**
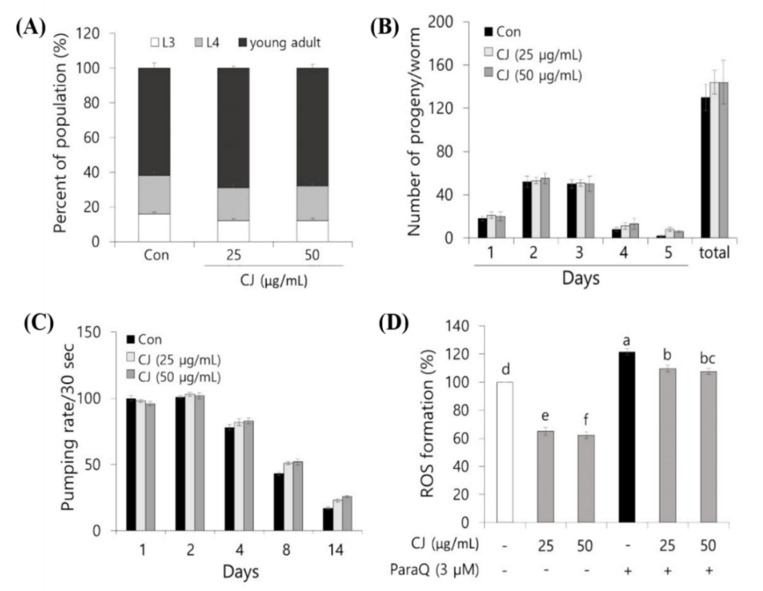
The effect of CJ extract on the growth factors and protective effect against oxidative stress of *C. elegans*. CJ extracts (25 and 50 μg/mL) does not change normal growth factors: (**A**) population, (**B**) progeny production. (**C**) CJ extracts (25 and 50 μg/mL) delayed the age-related behavioral symptom: pumping speed. Since the L1 stage, the nematodes were treated with 25 and 50 μg/mL CJ extracts for measuring normal growth factors and pumping speed. Each value represents mean ± standard error (n = 3 plates and 100 worms/plate for growth rate; n = 3 plates and five worms/plate for progeny production; n = 3 plates and 15 worms/plate for pumping rate) (**D**) CJ extracts (25 and 50 μg/mL) lowered ROS generation in *C. elegans*. Each bar represents the mean ± SD of three individual experimental results (n = 3). Letters above bars indicate significant differences according to Duncan’s multiple range test (*p* < 0.05).

**Figure 5 antioxidants-09-00200-f005:**
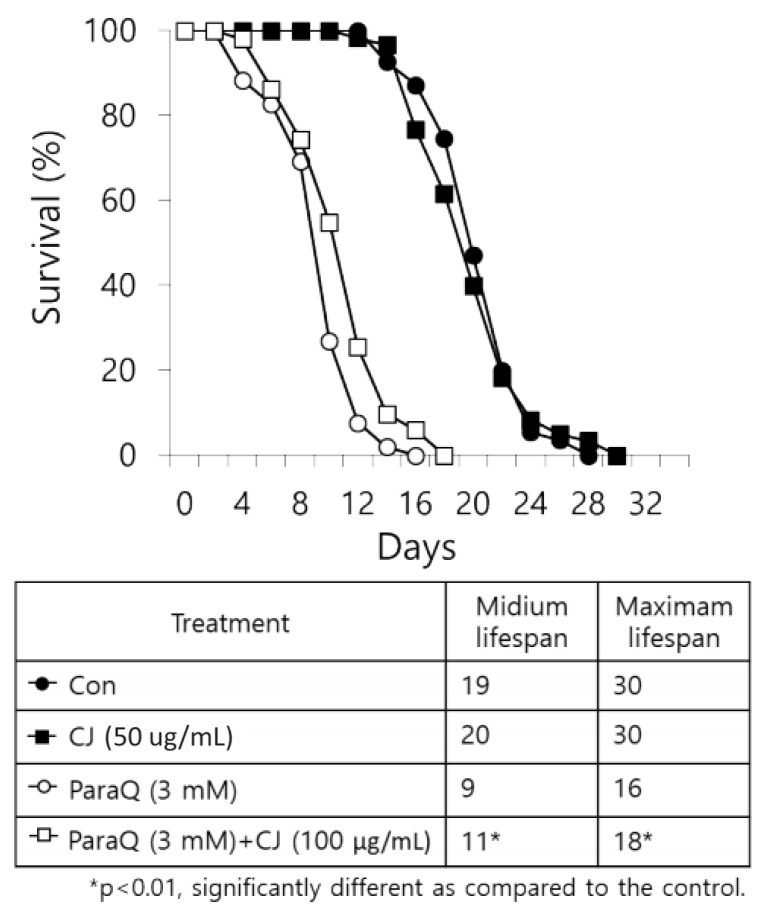
Survival curve of *C. elegans* treated with CJ extracts under normal and oxidative stress conditions. *C. elegans* were treated with CJ extracts (50 μg/mL) starting from young adult (day 0) and the survivals were recorded every other day until all the worms died. Oxidative stress was induced by 3 mM paraquat (paraQ). n = 3 plates and 100–120 worms/group.

**Table 1 antioxidants-09-00200-t001:** High performance liquid chromatography-electrospray ionization–mass spectrometry (HPLC–ESI–MS) analysis condition of phenolic compounds from *Cirsium japonicum*.

HPLC (UltiMate 3000, Thermo Scientific, CA, USA)	
Column Solvent Column temperature Wavelength Flow rate	Waters symmetry C18 column (Waters, 4.6 × 150 mm, 5 μm) (A) acetonitrile (B) 0.02% (*v/v*) aqueous phosphoric acid Gradient: 87% solvent B for 6 min, 85–87% solvent B for the next 3 min, 81–85% solvent B for 17 min, 72–81% solvent B for 28 min and a linear step from 72 to 87% solvent B for 12 min. 35 °C 340 nm 1.0 mL/min
**Mass spectrometer (Finnigan LCQ Deca XP plus, Thermo Scientific, CA, USA)**	
Polarity Capillary temperature Voltage Mass scan range (*m/z*)	ESI^+^ 275 °C Spray: 5 kV and capillary: 15 V 100–600

**Table 2 antioxidants-09-00200-t002:** Phenolic contents of 90 °C hot-air-, 70 °C hot-air-, shade-, and freeze-dried CJ.

Phenolic Compounds	Contents (mg/g DW)			
	90HCJ	70HCJ	SCJ	FCJ
Chlorogenic acid	3.51 ± 0.03 ^c^	13.35 ± 0.02 ^b^	22.14 ± 0.36 ^a^	21.84 ± 0.41 ^a^
Linarin	3.02 ± 0.12 ^cd^	3.24 ± 0.06 ^c^	4.11 ± 0.02 ^b^	6.96 ± 0.02 ^a^
Pectolinarin	0.68 ± 0.01 ^c^	0.82 ± 0.01 ^b^	0.94 ± 0.00 ^a^	0.96 ± 0.00 ^a^
Total	7.21 ^d^	17.41 ^c^	27.19 ^b^	29.76 ^a^

The values are mean ± standard deviation of three individual experimental results (n = 3). The superscripts a to d presented for each value in the same row represent statistically significant differences (*p* < 0.05). 90HCJ: hot-air-dried *C. japonicum* at 90 °C, 70HCJ: hot-air-dried *C. japonicum* at 70 °C SCJ: shade-dried *C. japonicum* and FCJ: freeze-dried *C. japonicum*.

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
