# Peer review of "Antioxidant Capacity of Thistle (Cirsium japonicum) in Various Drying Methods and their Protection Effect on Neuronal PC12 cells and Caenorhabditis elegans"

_antioxidants, 2020, doi:10.3390/antiox9030200_

Round 1

Reviewer 1 Report

In the manuscript titled:  “Evaluation of antioxidant capacities of thistle (Cirsium japonicum) dried by various drying methods and the protection effect of their extracts on neuronal PC12 cells and Caenorhabditis elegans against reactive oxygen species”, Authors investigated changes in phenolic compounds profile during different drying methods. They concluded that chlorogenic acid dominated in plyphenolic fraction and it was the most sensitive to high temperature during drying process. The authors conducted many in vitro tests in which they proved the anti-oxidative activity of thistle extract. Additionally they evaluated the protective effect on PC12 cells and Caenorhabditis elegans using the most active dry extract.

In my opinion the design of the experiment is correct and well documented. This work may contribute to a better understanding of the bioactivity of phenolic compounds, mainly chlorogenic acid and will be of interest to others in this field.

Minor corrections:

L184: citation 24 should be as [24], not as superscript

L225: citation 26 should be as [26], not as superscript

Figure 2: In the horizontal axis label there is ACJ abbreviation and should be SCJ

Author Response

Reviewer 1

In the manuscript titled:  “Evaluation of antioxidant capacities of thistle (Cirsium japonicum) dried by various drying methods and the protection effect of their extracts on neuronal PC12 cells and Caenorhabditis elegans against reactive oxygen species”, Authors investigated changes in phenolic compounds profile during different drying methods. They concluded that chlorogenic acid dominated in polyphenolic fraction and it was the most sensitive to high temperature during drying process. The authors conducted many in vitro tests in which they proved the anti-oxidative activity of thistle extract. Additionally they evaluated the protective effect on PC12 cells and Caenorhabditis elegans using the most active dry extract.

In my opinion the design of the experiment is correct and well documented. This work may contribute to a better understanding of the bioactivity of phenolic compounds, mainly chlorogenic acid and will be of interest to others in this field.

Minor corrections:

L184: citation 24 should be as [24], not as superscript

[Line 209]

The number of quotes has also been changed to match the order since the paragraph was reconstructed. We modified citation 27, which was marked as a superscript, to [27].

L225: citation 26 should be as [26], not as superscript

[Line 64]

The paragraph was moved to the introduction section and reconstructed for a clearer understanding.

The number of quotes has also been changed to match the order since the paragraph was reconstructed. Also, we modified citation 22, which was marked as a superscript, to [22].

Figure 2: In the horizontal axis label there is ACJ abbreviation and should be SCJ

ACJ was modified to be SCJ in Fig 2.

Thank you for your advice to be a better research article. We marked all revisions in red.

Reviewer 2 Report

Reviewer

Evaluation of Antioxidant Capacities of Thistle (Cirsium japonicum) Dried by Various Drying Methods and the Protection Effect of Their Extracts on Neuronal PC12 cells and Caenorhabditis elegans against Reactive Oxygen Species

The very ambitious overall aim of this work is to demonstrate that various drying methods (90°C hot-air-, 70°C hot-air-, shade-, and freeze- drying) can change the antioxidante activity of Thistle (Cirsium japonicum) extracts
and consequantly their protection effect on Neuronal PC12 cells and Caenorhabditis elegans. To this purpose, sets of analysis were done. I recognize that the authors provided a lot of work measuring and analyzing samples. This is a well organized work.  The data support the interpretations and conclusions.  The English usage is comprehensive but a few word choices could be improved.

Title: To me is too long, I suggest “Antioxidant Capacity of Thistle (Cirsium japonicum) in Various Drying Methods extracts and their Protection Effect on Neuronal PC12 cells and Caenorhabditis elegans

The KEYWORDSshould not be the same present in the tittle. If the authors change the title, they must put here ROS

Introduction

Some sentences must be rewritten and/or better contextualized (eg.: the first sentence should be the second one). The authors should talk a little about the importance of the major compounds contained in CJ. In general this section is clear but the English should be improved and the sentences should be re-written in order to make more sense.

Line 43: befor “first” is a dot??

Material and Methods

Why use the authors alcohol for the extraction?? Have they seen in the bibliography that is more efficient?? Why not methanol??

Results and Discussion

This is a very clear section. The authors discussed very well the results and support them.

Author Response

Reviewer 2

Evaluation of Antioxidant Capacities of Thistle (Cirsium japonicum) Dried by Various Drying Methods and the Protection Effect of Their Extracts on Neuronal PC12 cells and Caenorhabditis elegans against Reactive Oxygen Species

The very ambitious overall aim of this work is to demonstrate that various drying methods (90°C hot-air-, 70°C hot-air-, shade-, and freeze- drying) can change the antioxidante activity of Thistle (Cirsium japonicum) extracts and consequantly their protection effect on Neuronal PC12 cells and Caenorhabditis elegans. To this purpose, sets of analysis were done. I recognize that the authors provided a lot of work measuring and analyzing samples. This is a well organized work.  The data support the interpretations and conclusions.  The English usage is comprehensive but a few word choices could be improved.

Title: To me is too long, I suggest “Antioxidant Capacity of Thistle (Cirsium japonicum) in Various Drying Methods extracts and their Protection Effect on Neuronal PC12 cells and Caenorhabditis elegans

[Line 2-5]

We revised by referring your advice.

The KEYWORDS should not be the same present in the tittle. If the authors change the title, they must put here ROS

[Line 25-26]

We referred to the instruction of MDPI journal for submitting this manuscript. In the instruction, keywords are recommended to be specific to the article, yet reasonably common within the subject discipline. The keywords we used could present our article subject properly.

Introduction

Some sentences must be rewritten and/or better contextualized (eg.: the first sentence should be the second one). The authors should talk a little about the importance of the major compounds contained in CJ. In general this section is clear but the English should be improved and the sentences should be re-written in order to make more sense.

[Line 29-]

We reconstructed the paragraphs for a clearer understanding. And we corrected grammatical errors.

Line 43: befor “first” is a dot??

[Line 44]

We revised it to a dot.

Material and Methods

Why use the authors alcohol for the extraction?? Have they seen in the bibliography that is more efficient?? Why not methanol??

The use of methanol as an extracting solvent was allowed from 2008 in a study to develop materials for health-functional foods in Korea. But there is still consumer anxiety about the use of methanol in food development. To utilize thistle as a health-functional food, this study was conducted jointly with ‘Cheiljedang’ food company and we extracted thistle using ethanol to prevent any future anxiety.

Results and Discussion

This is a very clear section. The authors discussed very well the results and support them.

Thank you for your advice to be a better research article. We marked all revisions in red.

Reviewer 3 Report

The topic of research is interesting but some improvements are necessary. In the attach are all suggestions and needed corrections.

Obtained results are clearly accompanied by discussion. All results are well presented in graphs and tables.

Author Response

Reviewer 3

In ABSTRACT: please reorder the first sentence. From the existing one it is not clear what exactly the authors want to emphasize. My suggestion for the sentence: the aim of this study was to evaluate the phenol profile of thistle (Cirsium japonicum) by LC-ESI-MS, dried by different methods (90°C hot-air, 70°C hot-air, shade-, and freeze drying). Also, to evaluate the relationship between phenolic compounds content and antioxidant properties.

[Line 13-15]

We revised as you commented.

Key words: please avoid abbreviations, delete CJ from „thistle (Cirsium japonicum, CJ)“

[Line 25]

We removed ‘CJ’.

INTRODUCTION: „Thistle is characterized by leafs which has prickles on the edge.“ – please indicate which part of plant is used in medicinal purposes and in which kind of form are plant use (dried powder, extracts etc).

[Line 29-32]

We rewritten it with specific information.

In 2. MATERIALS AND METHODS; 2.1. Thistle (Cirsium japonicum) preparation and drying please indicate which part of plant was dried (the whole plant?). Also, before drying how the plants were prepared, washed, any soil impurities removed?

[Line 79-81]

We wrote the information specifiec preparation process.

Also, are the plants dried in elementary layer or?

We spread CJ thin on the shelves for the drying process. And, we regularly blended the thistles so that they could be evenly dried during the drying process.

….“CJ was completely dried“ – please indicate to which desired water content (%)

was plant material dried?

[Line 80-85]

We rewrote it with specific information: To value the relative effect of the drying methods, we measured the weight of fresh CJ and dried CJ using a moisture balance (MB45 moisture analyzer, Ohaus, USA). We assumed that once the ratio of the initial-final CJ weight after each drying process reached the measured value in the water analyzer, the CJ was completely dried.

Also, authors stated that hot-air drying was conducted in an oven – please instead oven, write laboratory drying chamber or laboratory dryer. Also, it is necessary to provide additional information about laboratory dryer – model type.

[Line 86]

We rewrote it with specific information: hot-air circulation oven (VB-200DL, Viosionbiotech, Gyeonggi, Korea)

The drying of CJ carried out by mentioned methods took a long time, especially in hot-air drying at 90 °C in which drying lasted for 2 days – at this relatively high temperature for drying of plant material, is the drying process carried constantly, respectively does the hot-air drying of CJ at 90 °C lasted even 48 h constantly? This question would not be necessary if the authors previously indicate which part of

plant was used for drying.

[Line 85-86]

We changed the unit of drying time from 'days' to 'hours' and rewritten it in more detail.

Authors indicate for freeze drying that this drying method was performed: „(d) Freeze-drying was conducted in a vacuum chamber at 0.5 mbar and -80ËšC in a freeze-dryer“ – but it is more accurate to state that freeze drying was conducted in freeze dryer at 0.5 mbar and -80 ËšC.

[Line 88-89]

We revised as you commented.

For preparing of CJ extracts – again, which part of plan tor the whole one, was used in extracts preparation? Also, how are the dry samples of the plant material milled before preparation of the extracts and if so to what particle size? How was the extraction method performed - extracted at 60°C for 6 h using 70% alcohol – in some kind of water bath or?

[Line 89-90]

All prepared CJ was extracted by reflux extraction and we revised it in the manuscript.

CONCLUSION – I suggest authors to emphasize does the prepared extract of CJ could be a potential product for use as strong antioxidant and for improvements in aging-related changes

[Line 423-424]

We presented in conclusion section that prepared CJ is a strong potential candidate for an antioxidant to delay aging.

Thank you for your advice to be a better research article. We marked all revisions in red.

Reviewer 4 Report

This study aimed at identifying the phenolic compounds of thistle dried with different methids and testing the antioxidant properties of these extracts. The phenolic compounds are decreased by drying at high temperatures, whereas they are maintained with low temperature drying methods. The samples keeping high phenolic compound content were also protective from oxidative stress in two biological systems, i.e. PC12 cells and the nematode C elegans.

The study is experimentally sound and well conducted, with an extensive amount of data; however the protective effects (on viability, membrane permeability, ROS generation in PC12; ROS generation and lifespan in C. elegans), though significant, are minor or marginal. The English text needs extensive improvement.

Author Response

Reviewer 4

This study aimed at identifying the phenolic compounds of thistle dried with different methids and testing the antioxidant properties of these extracts. The phenolic compounds are decreased by drying at high temperatures, whereas they are maintained with low temperature drying methods. The samples keeping high phenolic compound content were also protective from oxidative stress in two biological systems, i.e. PC12 cells and the nematode C elegans.

The study is experimentally sound and well conducted, with an extensive amount of data; however the protective effects (on viability, membrane permeability, ROS generation in PC12; ROS generation and lifespan in C. elegans), though significant, are minor or marginal. The English text needs extensive improvement.

We reconstructed the paragraphs for a clearer understanding. And we corrected grammatical errors.

Thank you for your advice to be a better research article. We marked all revisions in red.

Reviewer 5 Report

Major comments indicated by asterisk (*)

*Please reduce the length of the title and make the information within the title less specific so that the manuscript will appeal to a broader audience (e.g., neuronal PC12 cells is highly specific, drying methods should be reserved for the body of the manuscript, etc.).

* The paragraph organization, grammar (particularly noun-verb agreement) and spelling need to be improved throughout this manuscript to enhance clarity and add professionalism to the work.

* Examples of paragraphs that need to be reorganized/rewritten: ln 42-52, the logical transitions within this paragraph are missing; Ln58-62, this paragraph needs to be rewritten to better explain the overview and overall findings of this study. The current paragraph is poorly written and not well organized. The relevance of certain aspects of information presented in the paragraph (e.g., ‘according to drying methods of CJ’) is not well explained.

* Overall, the methods section needs to be modified to include details appropriately expressed to ensure that the experiments are reproducible. See below for some examples of recommendations (not meant to be all inclusive).

Ln 65, clarify Pocheon, South Korea

Ln 66, what was the method used for authentication of the plants? All that was mentioned was the organization that performed the method.

Ln 68, was the 90C drying performed in an oven?

Ln 69, what type of oven?

Ln 70, what was the ambient temperature of the shade drying?

Ln 71, what were the details of the vacuum chamber (company model)?

Ln 72, 70% (v/v) ethanol?

Overall, how were the CJ thistles layered in the various drying methods?

Ln 93 Table 1: mass tolerance?

Ln 96-98, This sentence should be removed. The sentence is unnecessary and too vague. The information is better described and referenced in the latter portion of this section.

Ln 99-100, this sentence needs to be reworded for clarity, as written it is difficult to tell what the authors are referring to.

Ln 106, need to define what ‘the mixture’ is referring to

Section 2.4 should be subdivided into separate sections based on the assay under description

Ln 130-121, please also clarify the final concentration of each component in the FRAP reagent

Ln 144, please clarify the organism source of the PC12 cells and define the details of PC12 cells upon first mention

Define all % within the methods (e.g., % by v/v or w/v)

Ln 149, how long does it generally take for the cells to spread out to an appropriate density

Ln 154, what was the volume of the PC12 cells at this point in the method? (so that the reader can define the ratio cell culture volume to the 0.2 ml of CJ extract added)

Ln 155, need citation with appropriate reference for the MTT assay

Ln 156, details on the stock of H2O2 (company, original concentration, age of solution) are needed to assess the validity of the 0.25 mM H2O2 claim

Ln 159-160, this portion of the methods is unclear - need define/quantify what is considered to be well placed and other aspects of this sentence.

Ln 160, define what the absorbance at 570 nm measure for this assay

Ln 182, OP50 refers to the strain of E. coli and, thus, the number should be placed after the E. coli species name.

Ln 184, the method used to corrode the nematode torsos with bleach should be presented in greater detail to ensure reproducibility

* Most of the protocols described in this study include a no treatment control (e.g., Lns 169, 175, 203, and others). This experimental approach should be modified to include a mock control. The mock control should include the delivery agent of the CJ extract as well as the treatment time. These two variables lack control in the current experimental design and will impact the results.

* The sample size and experimental reproducibility of most results are not clearly presented. See below for details.

Figure 1, no details on sample size, sample type (biological, technical) or experimental reproducibility

Table 2 and Figure 2, Experiments were performed in triplicate. Sample size and type (biological, technical) are not indicated, the lowercase letters (i.e., a-d) in the table/figure are not well defined - what is meant by significant differences? what is being compared? If the argument is that all of the conditions labeled a certain label are similar than that does not appear to be the case. For example, in figure 2, some of the conditions labeled [a] are similar to [b] than to [a]?

Figure 3, the authors indicate n=3 in the figure legend. However, the sample type (biological, technical) and experimental reproducibility are not indicated.

Figures 4 and 5, experimental reproducibility is not indicated.

* The conclusions should not be overstated. For example, Ln 222, the statement is made that drying is the most essential process for the storage of plants and is supported by citation of ref 27 (a study focused only on Phyllanthus amarus).  Thus, the claim that all plant types are best stored by drying seems unfounded.

* Sections of the manuscript need to be generally reorganized. For examples see the following:

Ln 222-227, should be moved to the introduction since this material is extensive and is based on past literature and not the results of this study.

Ln 228-230, this sentence would have been helpful to state at the end of the introduction and expand upon as this sentence more clearly explains what a portion of the study is about compared to the current paragraph that is written in ln 58-62.

Ln 238-253, this discussion should be reduced - too long when compared to the results presented in this section.

* Experimental reproducibility of the findings associated with the effects of different drying methods on the phenolic content (Ln 220-253) are not clarified. Thus, the significance of this section is difficult to assess. The differences observed in the levels of the three major phenolic compounds detected in the CJ extract (i.e., chlorogenic acid, linarin and pectrolinarin) are presumed to be due to different drying methods based on an apparent single experiment and one sample size per condition.

Ln 269: What do the DPPH, ABTS and ORAC values mean in terms of antioxidant?

Ln 303-304, micrograms per ml (please define what is measured to determine micrograms?)

Ln 308+, throughout manuscript be sure to use appropriate significant figure (e.g., 44% not 43.91%, 130% vs. 133.12%, etc.)

*A conclusion should be included at the end of this manuscript that clearly states what new knowledge is gained from this study that was not known from prior work.

Author Response

Reviewer 5

*Please reduce the length of the title and make the information within the title less specific so that the manuscript will appeal to a broader audience (e.g., neuronal PC12 cells is highly specific, drying methods should be reserved for the body of the manuscript, etc.).

[Line 2-5]

Because cell species used as nerve cells vary, many researchers use the cell name (e.g. PC12 cells) used in their respective studies in their titles It is also considered that 'various drying methods' are necessary for the title because the title should include the main content of the study. Therefore, the title was modified as follows: “Antioxidant Capacity of Thistle (Cirsium japonicum) in Various Drying Methods and their Protection Effect on Neuronal PC12 cells and Caenorhabditis elegans

* The paragraph organization, grammar (particularly noun-verb agreement) and spelling need to be improved throughout this manuscript to enhance clarity and add professionalism to the work.

We reconstructed the paragraphs for a clearer understanding. And we corrected grammatical errors.

* Examples of paragraphs that need to be reorganized/rewritten: ln 42-52, the logical transitions within this paragraph are missing; Ln58-62, this paragraph needs to be rewritten to better explain the overview and overall findings of this study. The current paragraph is poorly written and not well organized. The relevance of certain aspects of information presented in the paragraph (e.g., ‘according to drying methods of CJ’) is not well explained.

[Line 42-75]

We reconstructed the paragraphs and rewrote them for a clearer understanding.

* Overall, the methods section needs to be modified to include details appropriately expressed to ensure that the experiments are reproducible. See below for some examples of recommendations (not meant to be all inclusive).

Ln 65, clarify Pocheon, South Korea

[Line 78-79]

We revised in detail as you advised.

Ln 66, what was the method used for authentication of the plants? All that was mentioned was the organization that performed the method.

[Line 79]

The collected CJs were visually authenticated. We revised in the manuscript.

Ln 68, was the 90C drying performed in an oven? Ln 69, what type of oven?

[Line 86]

Hot-air circulation oven (VB-200DL, Viosionbiotech, Gyeonggi, Korea) was used for this research. We modified that part in detail.

Ln 70, what was the ambient temperature of the shade drying?

[Line 87]

The ambient temperature was controlled as 25.0±1.0 ËšC. We added the specific temperature in the manuscript.

Ln 71, what were the details of the vacuum chamber (company model)?

[Line 88-89]

The vacuum chamber is a part of freeze-dryer. We revised the sentence.

Ln 72, 70% (v/v) ethanol?

[Line 90]

We added “(v/v)”

Overall, how were the CJ thistles layered in the various drying methods?

We spread CJ thin on the shelves for the drying process. And, we regularly blended the thistles so that they could be evenly dried during the drying process.

Ln 93 Table 1: mass tolerance?

[Line 106]

The detected peaks from CJ extract were identified by comparing the retention time of the standard compounds. The mass spectra were then used to reaffirm whether the identified compounds were matched up the standard compounds used. Our purpose in mass analysis is to verify the molecular weight, not validation analysis. Therefore, the information given in Table 1 is considered sufficient.

Ln 96-98, This sentence should be removed. The sentence is unnecessary and too vague. The information is better described and referenced in the latter portion of this section.

The sentence has been removed and described later in the appropriate part to reflect your advice.

Ln 99-100, this sentence needs to be reworded for clarity, as written it is difficult to tell what the authors are referring to.

The sentence has been removed and described later in the appropriate part to reflect your advice.

Ln 106, need to define what ‘the mixture’ is referring to

That part has been removed as you advised.

Section 2.4 should be subdivided into separate sections based on the assay under description

[Line 110-158]

The portion was separated into the proper assay parts and explained under the newly moved parts.

Ln 130-121, please also clarify the final concentration of each component in the FRAP reagent

The FRAP solution is a ratio of 10:1:1 (v:v:v) of three individually prepared solutions (acetate buffer, ferric chloride, and TPTZ solution) manufactured at each appropriate concentration. The final concentration of each component of the FRAP reagent may vary depending on the ratio of mixing with the sample solution, and the ratio is adjusted by the experimenter according to the sample's condition. Therefore, it is reasonable to present the ratio when presenting the method, and many papers suggest only the ratio.

Ln 144, please clarify the organism source of the PC12 cells and define the details of PC12 cells upon first mention

[Line 163-164]

We added the specific information on the PC12 cells.

Define all % within the methods (e.g., % by v/v or w/v)

[Line  90, 166, 171, 181]

We revised it by referring to your advice.

Ln 149, how long does it generally take for the cells to spread out to an appropriate density

[Lin 168-169]

We revised the sentence in detail.

Ln 154, what was the volume of the PC12 cells at this point in the method? (so that the reader can define the ratio cell culture volume to the 0.2 ml of CJ extract added)

[Line 178]

For MTT assay, we added 200 μL of medium containing CJ extracts into the wells. That part was modified.

Ln 155, need citation with appropriate reference for the MTT assay

[Line 164]

To measure the effect of CJ on PC cells, we used our lab protocol and I mentioned on Line 163.

Ln 156, details on the stock of H2O2 (company, original concentration, age of solution) are needed to assess the validity of the 0.25 mM H2O2 claim

[Line 171-172]

We added specific information on H2O2.

Ln 159-160, this portion of the methods is unclear - need define/quantify what is considered to be well placed and other aspects of this sentence.

[Line 182-185]

We revised this part.

Ln 160, define what the absorbance at 570 nm measure for this assay

[Line 184-185]

We revised this part.

Ln 182, OP50 refers to the strain of E. coli and, thus, the number should be placed after the E. coli species name.

[Line 205]

OP50 was provided with nematode from Caenorhabditis Genetics Center. We added the information.

Ln 184, the method used to corrode the nematode torsos with bleach should be presented in greater detail to ensure reproducibility

[Line 209]

We revised it in detail.

* Most of the protocols described in this study include a no treatment control (e.g., Lns 169, 175, 203, and others). This experimental approach should be modified to include a mock control. The mock control should include the delivery agent of the CJ extract as well as the treatment time. These two variables lack control in the current experimental design and will impact the results.

[Line 186, 193, 200]

To experimental control, I treated DMSO (≤0.2%) in the negative groups. We revised those sentences to prevent misleading.

* The sample size and experimental reproducibility of most results are not clearly presented. See below for details.

Figure 1, no details on sample size, sample type (biological, technical) or experimental reproducibility

[Line 263, 266]

We indicated Figure 1 to show the peak separation for individual components in our HPLC analysis. Therefore, sample size (and replicate) was presented in Table 2.

Table 2 and Figure 2, Experiments were performed in triplicate. Sample size and type (biological, technical) are not indicated, the lowercase letters (i.e., a-d) in the table/figure are not well defined - what is meant by significant differences? what is being compared? If the argument is that all of the conditions labeled a certain label are similar than that does not appear to be the case. For example, in figure 2, some of the conditions labeled [a] are similar to [b] than to [a]?

[Line 266]

We revised Table 2 to reflect your advice.

[Line 304]

We modified Figure 2 to reflect your advice. And the letters presented above the same color bars represent statistically significant differences.

Figure 3, the authors indicate n=3 in the figure legend. However, the sample type (biological, technical) and experimental reproducibility are not indicated.

[Line 354]

We modified Figure 3 by referring to as your advice.

Figures 4 and 5, experimental reproducibility is not indicated.

We described the size of the experiment used for reproducibility in the Figure Legend as follows.

[Line 395, 413]

  • Figure 4A-4C: n=3 plates and 100 worms/plate for growth rate; n=3 plates and 5 worms/plate for progeny production; n=3 plates and 15 worms/plate for pumping rate
  • Figure 4D and Figure 5: We modified by referring to as your comment.

* The conclusions should not be overstated. For example, Ln 222, the statement is made that drying is the most essential process for the storage of plants and is supported by citation of ref 27 (a study focused only on Phyllanthus amarus).  Thus, the claim that all plant types are best stored by drying seems unfounded.

[Line 60-62]

The paragraph was moved to the introduction section and reconstructed for a clearer understanding.

We revised that part and add another reference to prevent overstate.

* Sections of the manuscript need to be generally reorganized. For examples see the following:

Ln 222-227, should be moved to the introduction since this material is extensive and is based on past literature and not the results of this study.

[Line 58-66]

The paragraph was moved to the introduction section and reconstructed for a clearer understanding.

Ln 228-230, this sentence would have been helpful to state at the end of the introduction and expand upon as this sentence more clearly explains what a portion of the study is about compared to the current paragraph that is written in ln 58-62.

[Line 67-75]

The sentence was moved to the introduction section and reconstructed for a clearer understanding.

Ln 238-253, this discussion should be reduced - too long when compared to the results presented in this section.

[Line 254-262]

We reduced that portion as you advised.

* Experimental reproducibility of the findings associated with the effects of different drying methods on the phenolic content (Ln 220-253) are not clarified. Thus, the significance of this section is difficult to assess. The differences observed in the levels of the three major phenolic compounds detected in the CJ extract (i.e., chlorogenic acid, linarin and pectrolinarin) are presumed to be due to different drying methods based on an apparent single experiment and one sample size per condition.

[Line 58-66]

The part is a description of the general drying effect and can support our research. However, it was not scientifically verified through our research results, so that part was moved to the introductory part.

Ln 269: What do the DPPH, ABTS and ORAC values mean in terms of antioxidant?

The meaning of the 'DPPH, ABTS, and ORAC' was explained on Line 272-276.

Ln 303-304, micrograms per ml (please define what is measured to determine micrograms?)

[Line 173-174]

We used the final concentration of CJ in a range of 0-50 μg extract per mL appropriately diluted for each experiment. We presented the explain in the Method section.

Ln 308+, throughout manuscript be sure to use appropriate significant figure (e.g., 44% not 43.91%, 130% vs. 133.12%, etc.)

[Line 323, 338, 33, 347, 376, etc.]

We revised as you advised.

*A conclusion should be included at the end of this manuscript that clearly states what new knowledge is gained from this study that was not known from prior work.

[Line 415-424]

We revised the conclusion section to reflect your advice.

Thank you for your advice to be a better research article. We marked all revisions in red.

Round 2

Reviewer 3 Report

The authors accepted all suggestions and correct the text so I recommend it for publishing

Author Response

Reviewer 3

The authors accepted all suggestions and correct the text so I recommend it for publishing

Thank you for your advice to be a better research article.

Reviewer 5 Report

The manuscript is much improved compared to the first draft but still contains statements that are not clearly presented to the reader.

For example, the no treatment control that the authors explain in the response to reviewers is a mock treatment with delivery agent (DMSO) is still not clearly stated in the methods. "The oxidative stress levels were expressed as a relative percentage of the control cells without CJ treatment." still implies that the control was simply no treatment.

Author Response

The manuscript is much improved compared to the first draft but still contains statements that are not clearly presented to the reader.

For example, the no treatment control that the authors explain in the response to reviewers is a mock treatment with delivery agent (DMSO) is still not clearly stated in the methods. "The oxidative stress levels were expressed as a relative percentage of the control cells without CJ treatment." still implies that the control was simply no treatment.

[Line 186-187 / 194-195 / 202-203]

We revised as you commented for a clear understanding.

Thank you for your advice to be a better research article. We marked the newly modified part with red letters and blue highlights.
